# Full-Length Model of SaCas9-sgRNA-DNA Complex in Cleavage State

**DOI:** 10.3390/ijms24021204

**Published:** 2023-01-07

**Authors:** Wenhao Du, Haixia Zhu, Jiaqiang Qian, Dongmei Xue, Sen Zheng, Qiang Huang

**Affiliations:** 1State Key Laboratory of Genetic Engineering, Shanghai Engineering Research Center of Industrial Microorganisms, MOE Engineering Research Center of Gene Technology, School of Life Sciences, Fudan University, Shanghai 200438, China; 2Multiscale Research Institute for Complex Systems, Fudan University, Shanghai 201203, China

**Keywords:** genome editing, CRISPR-Cas9, SaCas9, DNA cleavage, structure model

## Abstract

*Staphylococcus aureus* Cas9 (SaCas9) is a widely used genome editing tool. Understanding its molecular mechanisms of DNA cleavage could effectively guide the engineering optimization of this system. Here, we determined the first cryo-electron microscopy structure of the SaCas9-sgRNA-DNA ternary complex. This structure reveals that the HNH nuclease domain is tightly bound to the cleavage site of the target DNA strand, and is in close contact with the WED and REC domains. Moreover, it captures the complete structure of the sgRNA, including the previously unresolved stem-loop 2. Based on this structure, we build a full-length model for the ternary complex in cleavage state. This model enables identification of the residues for the interactions between the HNH domain and the WED and REC domains. Moreover, we found that the stem-loop 2 of the sgRNA tightly binds to the PI and RuvC domains and may also regulate the position shift of the RuvC domain. Further mutagenesis and molecular dynamics simulations supported the idea that the interactions of the HNH domain with the WED and REC domains play an important role in the DNA cleavage. Thus, this study provides new mechanistic insights into the DNA cleavage of SaCas9 and is also useful for guiding the future engineering of SaCas9-mediated gene editing systems.

## 1. Introduction

The CRISPR-Cas9 system has been emerging as a simple and powerful genome editing technology and is widely applied to the fields of gene therapy, drug screening, disease model construction, etc. [1,2,3,4]. This system mainly consists of two components: Cas9 endonuclease and single-guide RNA (sgRNA) [5,6]. Additionally, its DNA editing process is roughly as follows: firstly, Cas9 binds to sgRNA to form a ribonucleoprotein (RNP) complex; next, the RNP searches for the protospacer-adjacent motif (PAM) on DNA to locate the target site; meanwhile, ~20 nt guide sequence of the sgRNA tries to pair with the DNA sequences upstream of the PAM; once they are in a form of base pairing, Cas9 endonuclease is activated and then precisely cleaves the target site [7,8,9]. To date, the most widely used Cas9 proteins are *Streptococcus pyogenes* Cas9 (SpCas9) [10,11] and *Staphylococcus aureus* Cas9 (SaCas9) [12,13]. However, SpCas9 has 1368 amino acids (aa) (~4.1 kb), and its in vivo delivery with sgRNA and other accessory elements are usually limited by the adeno-associated virus (AAV) vector, of which packaging capacity is less than 4.7 kb [14]. In contrast, as seen in Figure 1A, SaCas9 has only 1053 aa (~3.1 kb) and is relatively smaller than SpCas9, and thus is easier to be delivered to the target cells [15]. So, SaCas9 is more suitable for therapeutic applications via AAV.

To facilitate the SaCas9 system more efficient, extensive studies have been devoted to understanding its molecular mechanisms to locate and cut the target DNA in the genome [16,17]. Among them, in vivo genome editing experiments have shown that SaCas9 could recognize the unique 5′-NNGRRT-3′ PAM sequence in DNA and found that when the length of the guide sequence increases from 20 to 21~23 nt, SaCas9 could achieve higher editing efficiency [13]. Additionally, two X-ray crystal structures of the SaCas9-sgRNA-DNA complex in the inactive state have been obtained (Figure 1B) [12]. These provide the structural basis for the PAM specificity and sgRNA recognition mechanisms of SaCas9. Later, another SaCas9 crystal structure showed new conformational changes of the SaCas9-sgRNA-dsDNA complex upon the AcrIIA14 binding [18]. All these results have deepened our understanding of the DNA cleavage mechanisms by the SaCas9 system and also enabled the design of SaCas9 variants with multiple-PAM recognition ability and high targeting specificity [19,20,21].

Although previous studies on the SaCas9-sgRNA-DNA ternary complex have offered valuable insights into its architecture and molecular basis, the active structure of SaCas9 similar to that of SpCas9 [7] remains unavailable. Additionally, in the current available structures of SaCas9, the catalytic center of the HNH domain is far away from the cleavage site of the target DNA strand, and the distance between the Cα atom of the catalytic residue H557 and the scissile P of the target strand is more than 35 Å [12,18]. Moreover, structural details essential for understanding the cleavage mechanisms of SaCas9 are missing. For example, the sgRNA stem-loop 2 that affects the cleavage activity has not been resolved, and the part of the non-target DNA strand bound to the catalytic center of the RuvC domain lacks. Since the active conformation is important for understanding the molecular mechanisms of the DNA cleavage and may provide a structural basis for system optimization, it is still interesting to obtain the active structure of SaCas9 in the DNA cleavage state.

In this study, we report the first cryo-electron microscopy (cryo-EM) map of the SaCas9-sgRNA-DNA ternary complex. This map reveals that the HNH nuclease domain is tightly bound to the cleavage site of the target DNA strand and closely interacts with the WED and REC domains. Moreover, it also captures the complete structure of the sgRNA, including the stem-loop 2 not resolved in the previous structures. Based on this map, we then build a full-length model for the ternary complex in the cleavage state by combining computational modelling. Additionally, this model enabled us to identify those residues for the interactions of the HNH domain with the WED and REC domains. Moreover, we found that the stem-loop 2 of the sgRNA tightly binds to the PI and RuvC domains and may also regulate the position shift of the RuvC domain, which leaves space for the activation conformational changes of the HNH domain. Site-directed mutagenesis and molecular dynamics (MD) simulations confirmed that the interactions of the HNH domain with the WED and REC domains are important for the DNA cleavage. Thus, this study provides new mechanistic insights into SaCas9.

## 2. Results

### 2.1. Cryo-EM Map of the SaCas9-sgRNA-DNA Complex

To obtain the SaCas9-sgRNA-DNA ternary complex, we first determined the components of the complex and tested their stability. Similar to our previous study [7], a 59 bp sequence from the *Fumarylacetoacetate hydrolase (Fah)* gene was selected as the target DNA (Figure 2A). As shown, its target sequence is 21 bp long, the PAM sequence is TTGAAT, and both ends are 16 bp long. Meanwhile, we used a sgRNA designed in the study by Ran et al. [13], with a 21 nt guide sequence at the 5′ end and a stem-loop 2 and 6-U at its 3′ end (Figure 2A). To promote the stable binding of the protein to DNA, we mutated D10 and N580 of the wild-type SaCas9 (wtSaCas9) into alanine and obtained dead SaCas9 (dSaCas9) (Appendix A) [12]. The electrophoretic mobility shift assay illustrated that dSaCas9 has no DNA cleavage ability (Appendix A, lane 3) but could cause the electrophoresis band of the substrate DNA to lag behind (Appendix A, lane 6), demonstrating that the RNA-guided dSaCas9 could stably bind to the substrate DNA. Therefore, to form the stable ternary complex, we co-incubated dSaCas9 with the sgRNA and the target DNA. We then used the 300 kV transmission electron microscope to image the samples and collect electron micrographs. Finally, 10,389 cryo-EM micrographs were collected, and an example is shown in Appendix A.

Next, to acquire a density map of the complex, we followed the single-particle 3D reconstruction workflow to process the obtained micrographs (Appendix A). Due to a large number of micrographs, we used the program PARSED developed by us [22] to pick complex particles from the micrographs (Appendix A). This program has been successfully applied to various cryo-EM datasets, including that of SpCas9 [7]. As a result, 3.87 million particles were picked from the micrographs. As can be seen in Appendix A, there is only one sharp peak in the mass distribution of the particle blobs calculated by PARSED, indicating that these particles are in good homogeneity. Then, we used them to perform multiple rounds of 2D classification and obtained those good particles with clear 2D class averaging images (Figure 2B). As seen, the 2D class averaging images clearly displayed the density projections of the ternary complexes (Figure 2B). Among them, both the 16 bp DNA helix structure downstream of the PAM (Figure 2B, arrow 1) and the hairpin structure of the sgRNA tetraloop (Figure 2B, arrow 2) are clearly visible.

Eventually, we extracted all the particles in the above 2D classes and performed unsupervised 3D classification and refinement (Appendix A). These resulted in a cryo-EM map at a resolution of ~4.3 Å, according to the standard of Fourier shell correlation (FSC) = 0.143 (Appendix A).

### 2.2. Conformational State of the Cryo-EM Density Map

To clearly define the structural features in the cryo-EM density map, we assessed the local resolution of the map and determined the position of each domain. As seen in Appendix A, most density regions belong to the high-resolution regions (<5.0 Å) [23], indicating that these domains in these regions are less flexible and heterogenous in conformations. To identify them, we performed rigid-body fitting with the complex structure of the inactive SaCas9 (PDB ID 5CZZ) and found that the corresponding fitted domains are REC, WED, PI, RuvC1, RuvC2, and a large part of RuvC3 (aa 650–716, 756–774; Figure 3A). Meanwhile, those nucleic acids bound to SaCas9 could also be fitted into these density regions, namely the PAM double helix and the RNA: DNA heteroduplex (Figure 3A). In addition, the density regions for the sgRNA scaffold are clear too, including the stem-loop 2 essential for in vivo activity. In particular, we found that the stem-loop 2 is in close contact with the densities of the PI and RuvC domains, with a hairpin-shaped structure (Figure 3A). Hence, most parts of the SaCas9-sgRNA-DNA complex are distributed in the high-resolution regions of the map, except for the HNH domain, L1/L2 linkers, and a small part of RuvC3 (aa 717–755). This certainly provides a solid experimental basis for building a reliable full-length model for the ternary complex.

Next, we analyzed the density regions for the HNH domain and the L1/L2 linkers, which are more flexible in the DNA cleavage process [24]. As indicated in Figure 3B, at the contour level of ~0.0022, there exists a special density region whose position is different from that of the HNH domain in the inactive structure (PDB ID 5CZZ). The HNH domain in 5CZZ is close to the RuvC domain, and its catalytic center is far from the target DNA cleavage site (Figure 3B) [12]. In contrast, the mentioned density region in our map is tightly bound to that of the target DNA strand and also contacts the densities of the WED and REC domains (Figure 3B). We found that the position of this region is very similar to that of the HNH domain of SpCas9 in the cleavage state (Figure 3B) [7]. No doubt, this density region corresponds to the HNH domain of SaCas9. Unfortunately, no clear density was presented for two L1/L2 linkers, suggesting again that the conformations of these two segments are flexible in that structure. This is consistent with that in the activation state, where they undergo helix-to-loop transition when the HNH domain transfers from the inactive to the active states. Thus, our cryo-EM density map captured a new structure of SaCas9 with an HNH conformation different from that of the inactive structure 5CZZ. Similar to that of SpCas9, the HNH domain in this structure appears to be in the cleavage activation state.

To offer insights into the molecular mechanisms of the DNA cleavage, we built a full-length model for the SaCas9 in complex with sgRNA and DNA based on the cryo-EM structure. First, according to the experiments, we directly constructed models for the domains in the high-resolution regions of the cryo-EM density map, as described in Materials and Methods. As expected, these domains match the experimental densities very well, especially the helical segments (Figure 4A). So, the models for these segments are fully supported by the experiment data and could be considered as a stable scaffold of the complex. Certain regions, such as HNH and L1/L2, have relatively lower resolutions, which are not sufficient to support our reliance on cryo-EM densities exclusively for model construction (Appendix A). We predicted the models according to their density profiles and superimposed positions with the reference structures (PDB ID 5Y36, 7EL1, Figure 4B). Additionally, we then corrected and optimized them by computational modelling. As seen, the built model of the HNH domain is well wrapped in the density at the contour level of ~0.0022 (Figure 4C). Finally, we obtained a full-length model of the SpCas9-sgRNA-DNA ternary complex by assembling an electron microscopy-based atomic model stable scaffold with a computationally optimized prediction model. The components of the complex model are SaCas9 (aa 2-1053; D10A/N580A), 104 nt sgRNA, 40 nt target DNA strand, and 28 nt non-target DNA strand (Figure 5A and Appendix A).

### 2.3. Model Features of the SaCas9-sgRNA-DNA Complex in Cleavage State

As seen in Figure 5B, comparing to 5CZZ, the full-length model shows that the HNH domain, as a whole, flips toward the cleavage site of the target strand, and the distance between the Cα atom of the catalytic residue H557 and the scissile P reduces from ~46.2 Å to ~11.5 Å (Figure 5B). Again, these strongly support that the HNH domain in the model approaches a state that is competent for the DNA cleavage. Meanwhile, as the linker domain of HNH, L1 changes from the helical to a more flexible loop-like conformation, and the L2 domain also docks to the PAM from a position close to the end of heteroduplex (Figure 5C). In addition, the model shows that the cryo-EM density contacts of the HNH domain with the REC and WED domains are attributed to the hydrogen bonding between the domains: S581 of the HNH domain with E131 of the REC domain (Figure 5D), and S595 and R586 of the HNH domain with K811 and D812 of the WED domain, respectively (Figure 5E). Since previous studies only showed that the REC and WED domains are responsible for stabilizing the sgRNA backbone and recognizing DNA [12], this new finding suggests that both the REC and WED domains may also mediate the cleavage-activating process of the HNH domain.

Very interestingly, as seen in Figure 6A, the catalytic site of RuvC domain in 5CZZ is covered by the HNH and L1 domains. However, in our model, with the repositioning of the HNH, L1, and L2 domains, the catalytic site is exposed, and they provide a structural channel for the non-target DNA strand to approach the catalytic site (Figure 6A). The model shows that the non-target DNA strand fragment of the PAM upstream stably inserts into the positively charged groove formed by the HNH, L2, RuvC, and PI domains (Figure 6B). This electrostatic groove may stabilize the non-target DNA strand and promote the catalytic contact of the active center of the RuvC domain with the non-target DNA cleavage site. In fact, the cleavage site in our model is very close to the catalytic residue D10, and the distance between the scissile P and its Cα atom is about 11.5 Å (Figure 6C).

Significantly, the full-length model also showed the hairpin structure of the stem-loop 2 in the sgRNA and revealed that the stem-loop 2 binds tightly to the positively charged groove formed by the PI and RuvC II domains (Figure 7A). Moreover, we found that compared to 5CZZ, the RuvC domain has a large shift towards the stem-loop 2 (Figure 7B). This indicates that stem-loop 2 may also regulate the position shift of the RuvC domain, which could leave space for the binding of the substrate DNA and the conformational changes of the HNH domain. In addition, the structure of the nucleic acid 6-U on the 3′ tail of sgRNA has no stable density and its predicted model appears flexible, implying that it might not be an essential sgRNA component for DNA cleavage (Figure 7B). In addition, we observed that the double helix in the downstream of PAM is also embedded in the positively charged groove of the PI domain (Figure 7C), and the phosphate backbone of dA15 in the target DNA strand binds to K1023 with a hydrogen bond (Figure 7D). Consistently, related dynamics studies on SaCas9 also indicated the existence of this post-PAM interaction site [17]. Moreover, a similar DNA interaction site has also been detected for SpCas9 [7,25], highlighting the important role of post-PAM interactions in mediating substrate binding for Cas9.

### 2.4. Site-Directed Mutagenesis and Molecular Dynamics Simulations

As mentioned, our structure has revealed new conformational rearrangement of the HNH domain and interactions of the HNH domain with the REC and WED domains. To verify whether these interactions mediate the cleavage-activating process of the HNH domain, we mutated the key residues E131 of the REC domain and K811 and D812 of the WED domain to alanine, respectively, and then detected in vitro cleavage activities of these mutants (E131A, K811A, and D812A). Compared with wtSaCas9 with a cleavage rate of ~53%, both K811A and D812A increase the DNA cleavage activity to ~67% and 69%, respectively (Figure 8A). For the E131A, the catalytic rate is ~46%, slightly lower than that of wtSaCas9 (Figure 8A).

Meanwhile, we noticed that mutations in the WED domain (K811A and D812A) and in the REC domain (E131A) have different effects on the cleavage activity. These mutations in two different domains may cause the HNH domain to change its position close to or away from the DNA cleavage site, which, in turn, have different effects on the activity. To test this hypothesis, we constructed the models for wtSaCas9 and these mutants, and then performed about 500 ns MD simulations. As indicated by the RMSDs in Appendix A, after the simulation time of 200 ns, the systems were equilibrated. Then, we selected the MD trajectories between 200–500 ns to analyze the distance between the catalytic residue H557 and the scissile P (D_557-P_; Figure 5B).

As shown in Figure 8B, the D_557-P_ of wtSaCas9 fluctuates in the range from 10.1 to 15.7 Å, and there is only one distribution peak (D_557-P_ = 13.2 Å). This indicates that the HNH domain is always in close contact with the cleavage site at the given time. For mutants D812A and K811A with increased activity, their D_557-P_ distributions are more concentrated, and their distribution peaks are closer to the cleavage site than that of wtSaCas9 at 11.7 Å and 12.2 Å, respectively (Figure 8B). Likely, weakening the interactions between the WED and HNH domains leads the HNH domain to the cleavage site. However, for E131A with decreased activity, the D_557-P_ distribution is more decentralized than that of wtSaCas9, in the range of 9.7~17.4 Å, and although the D_557-P_ distribution peak is 12.4 Å, its density is relatively lower than that of wtSaCas9 (Figure 8B). So, reducing the interactions between the REC and HNH domains could decrease the chance of the HNH domain contacting the cleavage site. Taken together, consistent with the results of the site-directed mutagenesis, the MD simulations also supported that the interactions of the REC and WED domains with the HNH domain mediate the cleavage activation of the HNH domain.

## 3. Discussion

In this article, we obtained a cryo-EM map of the SaCas9-sgRNA-DNA ternary complex at 4.3 Å. It reveals that the HNH nuclease domain is in close contact with the cleavage site of the target DNA strand, and closely interacts with the WED and REC domains too. Meanwhile, it shows the structural orientation of the non-target DNA strand upstream of the PAM and captures the complete structure of the sgRNA. Based on this density information and computational modelling, we built a full-length model for the ternary complex in cleavage state. Our model shows that the active site of the HNH domain is about 11.5 Å away from the cleavage site of the target DNA strand. Moreover, we found that the stem-loop 2 of the sgRNA tightly binds to the PI and RuvC domains, and it may also regulate the position shift of the RuvC domain, which leaves space for the activation conformational changes of the HNH domain. Further mutagenesis and MD simulations supported that the interactions of the HNH domain with the WED and REC domains mediate the DNA cleavage.

By determining the cryo-EM density of the SaCas9-sgRNA-DNA complex, we have characterized the cleavage state of the complex in which the HNH domain contacts the cleavage site of the target DNA strand. Compared to that of SpCas9 [7], this structure has a higher resolution and more clearly shows the conformation changes of the HNH domain approaching the active state from the inactive state. In comparison with the inactive structures of SaCas9 [12,18], our structure presents the position of the non-target DNA strand upstream of PAM and the complete sgRNA. Moreover, it reveals how the HNH domain interacts with the REC and WED domains in the cleavage state of SaCas9 and indicates their important roles in regulating the DNA cleavage of SaCas9. So, structural information from this study could expand our understanding of the RNA-guided SaCas9 molecular mechanisms for the recognition and cleavage of DNA and may also offer guidance for the design or optimization of high-fidelity variants of SaCas9 [26], including the rational engineering of the SaCas9-mediated base editor [27], etc. In addition, the obtained structure might serve as a structural template for the homologous nucleases of SaCas9 to build their structural models in the active state [28].

Due to the dynamic nature of the SaCas9 protein itself, the atomic resolution of its transient catalytic structure is very difficult to be obtained. In addition, structural details of the relatively loose and flexible regions in the complex were not clearly captured yet, such as those of the L1 and L2 domains. However, dynamics studies have suggested that both domains might play a role in promoting the conformational changes of SaCas9 [29]. Therefore, further structural investigations are needed to clarify such detailed molecular mechanisms. Indeed, using our full-length model it is also possible to carry out long time scale MD simulations to investigate the dynamic process of SaCas9 from the inactive state to the active state, as in previous studies on SpCas9 [30].

In conclusion, because of its small size, SaCas9 stands out among many Cas9s and has been involved in more developments for therapeutic applications [31]. Many engineering efforts are being taken to address its functional deficiencies, such as the off-target effects and low efficiency, etc.; and structure-based optimization is particularly important for these efforts. So, in this work we have solved the first cryo-EM structure of the SaCas9-sgRNA-DNA ternary complex. This structure enabled us to build the full-length model of the ternary complex in the cleavage state and to reveal that the interactions of the HNH domain with the WED and REC domains mediate the DNA cleavage. In the future, our results may be used to guide the design of high-fidelity variants of SaCas9 and the optimization of SaCas9-mediated base editor, etc. Thus, our study not only provides new mechanistic insights into the DNA cleavage by CRISPR-Cas9 but also offers useful guidance for engineering SaCas9-based gene-editing systems.

## 4. Materials and Methods

### 4.1. Protein Expression and Purification

The sequence encoding the wide-type SaCas9 (wtSaCas9) was cloned into the pET-28a expression plasmid, and the N-terminal of the protein sequence was fused with a His6 tag. The other mutant plasmids were obtained by PCR-mediated site-directed mutagenesis of the wtSaCas9 plasmid. The primer sequences for constructing these plasmids are listed in Appendix A.

The activity-dead SaCas9 (dSaCas9) protein in the cryo-EM experimental sample was expressed and purified by Novoprotein Co., Ltd. The plasmid was expressed in E. coli BL21 (DE3). Briefly, cells were grown in LB medium at 37 °C until the OD600 = 0.6 ~ 0.8 and were induced with 0.5 mM IPTG. Growth continued at 16 °C overnight, and the cells were harvested by centrifugation, resuspended in lysis buffer (20 mM Hepes, 300 mM KCl, 10 mM MgCl_2_, 1 mM DTT, 1% Sucrose, 2 mM EDTA, pH 7.5), and lysed by sonication on ice. After centrifugation at 11,000 rpm for 1 h, the supernatant was filtered with the 0.2 μm membrane. Finally, the protein solution was successively purified by the cOmplete His-tag column and the Phenyl-HP column. The protein expression process of the wtSaCas9 and other mutants were the same as that of the dSaCas9. The proteins were purified by a Ni-NTA Sepharose (Qiagen) column on the BioLogic LP system (Bio-Rad). The purity of all the protein was verified by SDS-PAGE analysis; the results are shown in Appendix A. The purified proteins were concentrated to 1~2 mg/mL in the buffer (20 mM Hepes, 500 mM KCl, 1 mM DTT, pH 7.5) and stored at −80°C.

### 4.2. Preparation of Target DNA and sgRNA

The 59 nt target/non-target DNA strand used in the cryo-EM complex sample was synthesized by Sangon Biotech Co., Ltd., Shanghai, China. Two DNA strands with the same molar concentrations were mixed in the NEB buffer 2.1. The mixture was heated to 95 °C for 3 min, then was slowly cooled to 25 °C to form a 59 bp target DNA. The 1000 bp substrate DNA contains the same target sequence and PAM as the 59 bp target DNA. It was amplified by PCR using Pfu DNA Polymerase (TIANGEN, Beijing, China), and was purified using the AxyPrep TM DNA Gel Extraction Kit (Axygen Biotechnology, Taizhou, China). The 105 nt sgRNA was transcribed using the MEGAshortscript T7 Transcription Kit using DNA template and purified using the MEGAclear Transcription Clean-Up Kit (Thermo Fisher Scientific Co., Ltd., Waltham, MA, USA). The primer sequences used for nucleic acid sequences preparation are listed in Appendix A.

### 4.3. Cryo-EM Samples and Data Collection

The preparation of the SaCas9-sgRNA-DNA ternary complex sample was divided into three steps: first, the dSaCas9 protein was mixed with the 105 nt sgRNA in the reaction buffer (20 mM Tris-Cl, 100 mM KCl, 5 mM MgCl_2_, 1 mM DTT, pH 7.5) and incubated at 37 °C for 30 min. Then, the 59 bp target DNA was added to the solution and incubated at 37 °C for 1 h to obtain the sample solution; after mixing the components, the sample solution was incubated at 18 °C overnight to disperse the complexes. The dSaCas9, sgRNA, and target DNA were mixed at a molar ratio of 1:1.2:1.3 in the sample solution, and their corresponding concentrations were 1.75, 2.10 and 2.28 μM, respectively. The formation of the ternary complex was confirmed by agarose gel electrophoresis.

The cryo-EM grid samples were prepared by the standard rapid freezing procedure of the Vitrobot (FEI) system. First, ~2.5 μL dispersed complexes solution was loaded on the glow-discharged holey carbon grid (Quantifoil, R1.2/1.3, 200 mesh). Then, the grid samples were blotted for 4 s in a 16 °C and 100% humidity environment. Finally, it was quickly inserted into the liquid ethane to form the frozen samples. The Titan Krios G3i EM was used to image the frozen samples at 300 kV, and the Gatan K3 Bioquantum direct electron detector was used for the micrograph collection. The micrographs were recorded at a pixel size of 0.85 Å. The total exposure dose was 38 e/Å^−2^, which was fractionated into 38 sub-frames, and each of the images was recorded with random defocus in the range of −1.8 to −2.6 μm. Detailed processing parameters are shown in Appendix A.

### 4.4. Single Particle 3D Reconstruction

All collected cryo-EM micrographs (10,389) were aligned for motion correction [32], and the sub-frames of each image were superimposed into a single micrograph. As shown in Appendix A, the program PARSED [22] was used to select all complex particles from the micrographs. About 3,872,337 particles were picked and then used for the single-particle 3D reconstruction by RELION (Version 3.0.8) [33]. First, all particles were divided into 10 groups, and each group performed at least 5 rounds of 2D classifications; next, about 757,802 particles in the clear 2D classes were selected and fed into unsupervised 3D classification. After that, those particles with the highest resolution density map were selected for the next 3D refinement and post-process optimization. With a mask of initial binarization threshold = 0.0025 and an automated B-factor, the post-processing was performed to sharpen the structure to a resolution of 4.3 Å, according to the gold-standard FSC = 0.143.

### 4.5. Full-Length Model Building of the Complex

The procedures to build the full-length model of the complex were divided into three phases, as described in Appendix A. First, for all the SaCas9 domains and nucleic acid segments in the high-resolution regions of the density map, their models were constructed using corresponding parts of the crystal structure 5CZZ [12] as the templates and then refined by flexible fitting based on the cryo-EM densities. Second, those domains in the low-resolution regions were built and then assembled with the first part of the complex by combining structural prediction and manual placement guided by the densities. Finally, the whole complex was refined by molecular dynamics simulations, to optimize the atomic positions of the domains in the low-resolution regions.

Using 5CZZ as the template, the initial models for the SaCas9 domains in the high-resolution regions were built by homology model with MODELLER, and only the conformation of a small part of RuvC3 (aa 717–756) was predicted by AlphaFold2 using the sequence of RuvC3 (aa 650–774) as the input. The sgRNA and most bases of the target DNA were built by base substitution with the template from 5CZZ. The missing structures of sgRNA stem-loop 2 and 6-U were constructed with the RNA Assembly module of Rosetta (Ver. 3.6) [34,35]; other missing bases of two DNA strands were built using the templates from the SpCas9 structure (PDB ID 5Y36). Eventually, the partial SaCas9, sgRNA, and two DNA strands were assembled into a complex structure. We conducted rigid fitting to fit this structure into the cryo-EM density map with Situs (Ver. 3.1) [36], and then flexible fitting to refine it with Rosetta (Ver. 3.6) [37].

Next, we connected the models of the remaining L1/L2 linkers and the HNH domain in the low-resolution regions to the above-refined structure, which could be considered as a stable scaffold. Since the experimental densities are not enough to accurately determine the residue positions, the HNH structure was directly adopted from 5CZZ, and the L1/L2 conformations were predicted and manually checked by referring to those in the structure of 7EL1 [18]. Three structures were then manually combined with the scaffold structure according to the HNH position in the density map, as well as the HNH position in the active structure of SpCas9 [7] superimposed with the scaffold. Finally, the full-length model of the whole SaCas9-sgRNA-DNA complex was optimized by molecular dynamics simulation with GROMACS (Version 5.1.4) and then validated with PHENIX (Ver. 1.19.2) [38].

### 4.6. Activity Detection of SaCas9 Proteins

The wtSaCas9/SaCas9 mutant proteins (~100 nM) were first mixed with sgRNA (~200 nM) in the reaction buffer (20 mM Hepes, 100 mM KCl, 1 mM DTT, 0.5 mM EDTA, 2 mM MgCl_2_, 5% glycerol, pH 7.5) and incubated at 37 °C for 30 min. Then, the substrate DNA (~40 nM) was added to the Cas9-sgRNA solution and incubated at 37 °C for 1 h. Finally, the cleavage products were detected by gel electrophoresis on 1% agarose gel stained with GeneGreen nucleic acid dye (Tiangen Biotech Co. Ltd., Beijing, China). The activity was quantified using ImageJ, with the percentage of the cleavage rate being $100\times \left(1-\sqrt{1-fraction\;cleavaged}\right)$ [39].

### 4.7. Molecular Dynamics Simulations

GROMACS (Version 5.1.4) was used to perform unbiased explicit solvent molecular dynamics simulations on the ternary complex of wtSaCas9 and its mutants. The Amber99SB-ILDN force field [40] and the TIP3P water model [41] were employed. The all-atom structure of the complex was placed in the center of a triclinic water box with a size of 92.4 × 116.4 × 127.7 Å. Certain numbers of Na^+^ and Cl^-^ ions were added to the systems for setting an ionic concentration of 150 mM and neutralizing the systems. The periodic boundary conditions were used in the simulations. The particle mesh Ewald (PME) method [42] was used to calculate the electrostatics. A cut-off distance of 14 Å was used for the van der Waals interactions, and the LINCS algorithm [43] was used to constrain all the bond lengths and angles. The integration step was 2 fs. First, the systems were minimized with the steepest descent algorithm until the maximum force < 1000 kJ·mol^−1^·nm^−1^. Next, the systems were equilibrated in the NVT ensemble for 100 ps at 310 K controlled by the V-rescale method [44]. Then, an NPT equilibration for 100 ps was followed by setting the system pressure to 1 bar using the Parrinello–Rahman barostat [45]. Finally, unbiased production simulations were performed for more than 100 ns.

## Figures and Tables

**Figure 1 ijms-24-01204-f001:**
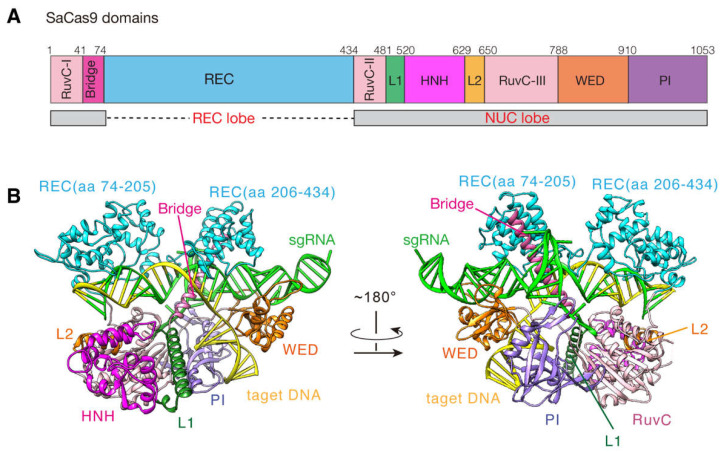
The structural domains of the SaCas9-sgRNA-target DNA complex. (**A**) Domain organization of SaCas9. (**B**) Ribbon representations of the complex structure in inactive state (PDB ID 5CZZ). The REC, RuvC-I~III, HNH, L1, L2, WED, PI and Bridge domains are colored in cyan, light pink, pink, dark green, orange, brown, purple and rose red, respectively. The sgRNA is colored in green; the target DNA is colored in yellow.

**Figure 2 ijms-24-01204-f002:**
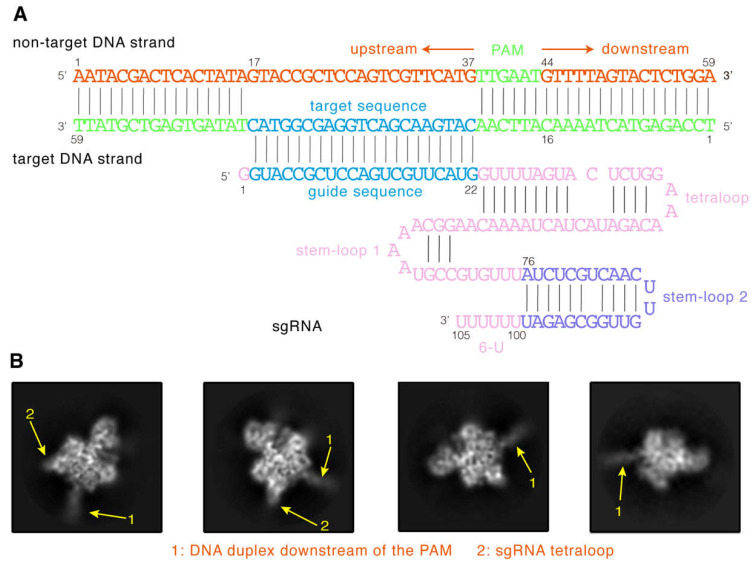
The cryo-EM structure of the ternary complex. (**A**) Schematic diagrams of 105 nt sgRNA and 59 bp target DNA of the ternary complex. (**B**) Representative 2D classes of the complex particles. The positions of the substrate DNA downstream of the PAM are indicated by arrows 1, and the positions of the sgRNA tetraloop are indicated by arrows 2.

**Figure 3 ijms-24-01204-f003:**
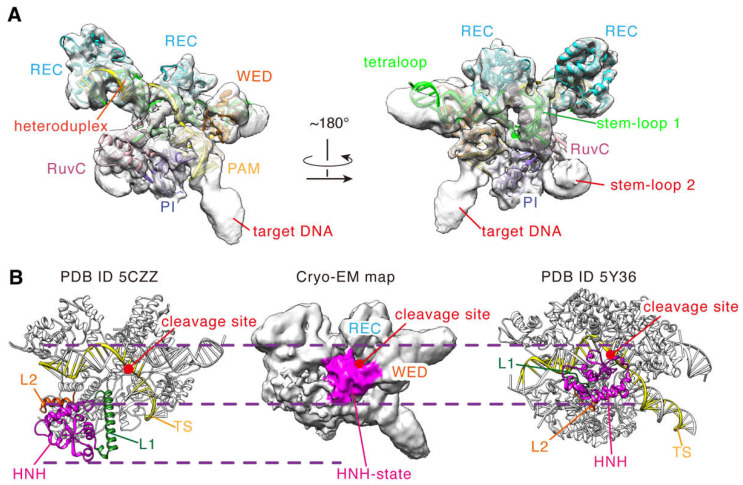
The conformational state of the cryo-EM density map. (**A**) Rigid-body fitting of the crystal structure (PDB ID 5CZZ) with the cryo-EM density map (contour level = ~0.0047). The structure of 5CZZ is shown by cartoon. (**B**) Comparison of the conformational states of the HNH domain in the density map (contour level = ~0.0022) with those of PDB ID 5CZZ and 5Y36. The dashed lines indicate the boundaries of the HNH domains.

**Figure 4 ijms-24-01204-f004:**
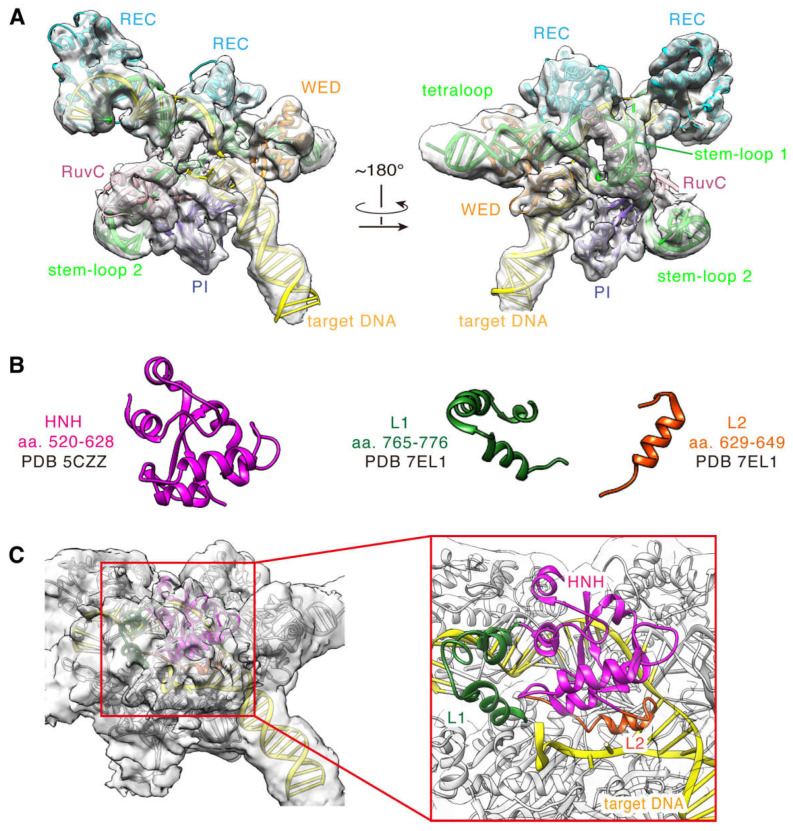
Cryo-EM structure of SaCas9-sgRNA-DNA ternary complex. (**A**) The EM-based structure with its cryo-EM density map (contour level= ~0.0047). (**B**) The conformation of the reference structure for HNH (pink), L1 (dark green), and L2 (orange) domains. (**C**) The HNH/L1/L2 domain model with its cryo-EM density map (contour level = ~0.0022). The HNH, L1, and L2 domains are represented by cartoons in pink, dark green, and orange, respectively.

**Figure 5 ijms-24-01204-f005:**
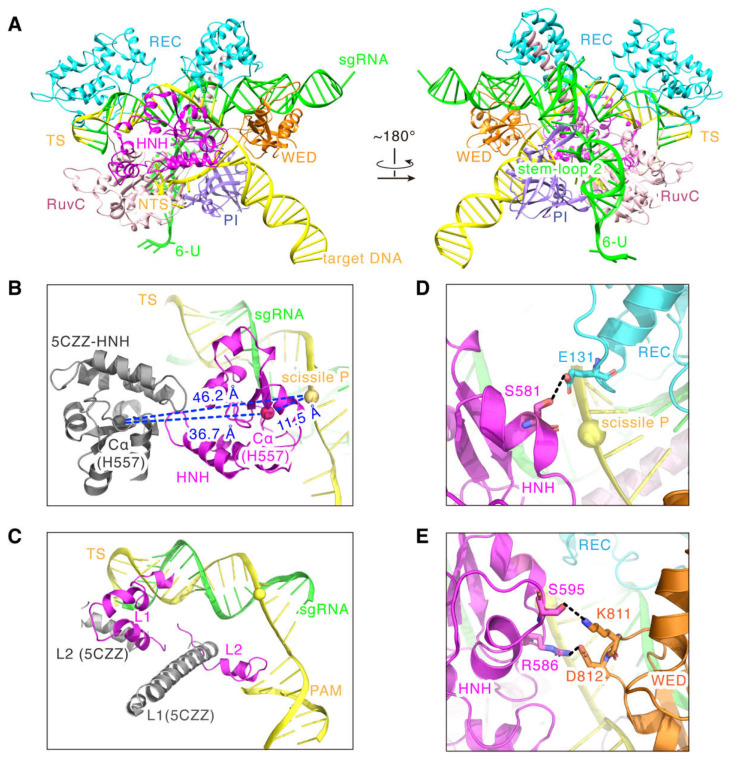
The full-length model for the SaCas9-sgRNA-DNA ternary complex. (**A**) The full-length model of SaCas9 (aa 2-1053; N580A, D10A) in complex with sgRNA (104 nt), target DNA strand (40 nt) and non-target DNA strand (28 nt). (**B**) The HNH domain states in PDB 5CZZ (gray) and the full-length model (pink), respectively. (**C**) The L1 and L2 states in PDB 5CZZ (gray) and the full-length model (pink), respectively. (**D**) The close contacts of the HNH domain (pink) with the REC domain (cyan). Hydrogen bonds are shown as dotted lines. (**E**) The close contacts of the HNH domain (pink) with the WED domain (brown). Hydrogen bonds are shown as dotted lines.

**Figure 6 ijms-24-01204-f006:**
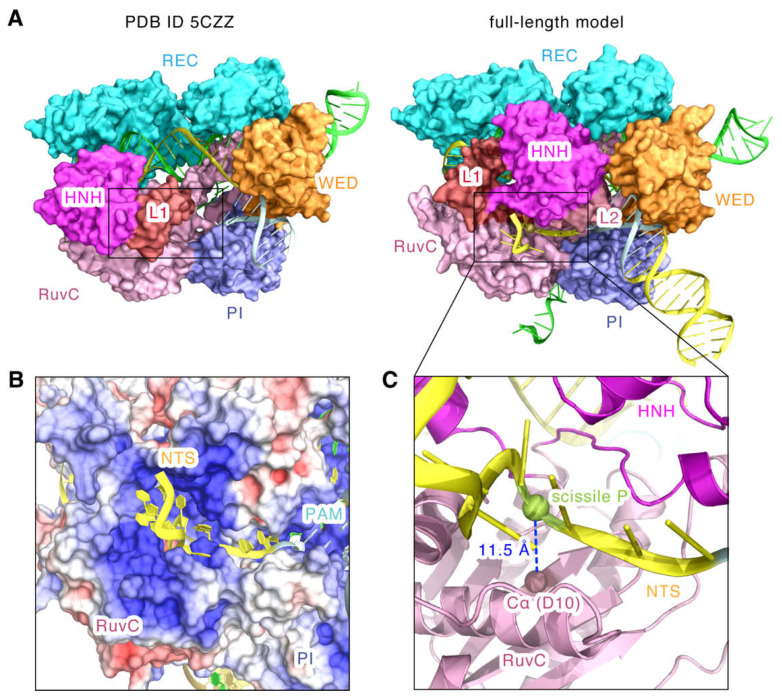
The topology of the non-target DNA strand bound to the RuvC domain. (**A**) Structural comparison of the SaCas9 NUC lobes in PDB ID 5CZZ and the full-length model. Position of the non-target DNA strand is indicated by a black box. (**B**) The electrostatic surface potential of SaCas9 around the non-target DNA strand. (**C**) Distances from the Cα atom of the catalytic residue D10 of the RuvC domain to the phosphorus of the scissile bond. The non-target DNA strand is represented by the cartoon in yellow.

**Figure 7 ijms-24-01204-f007:**
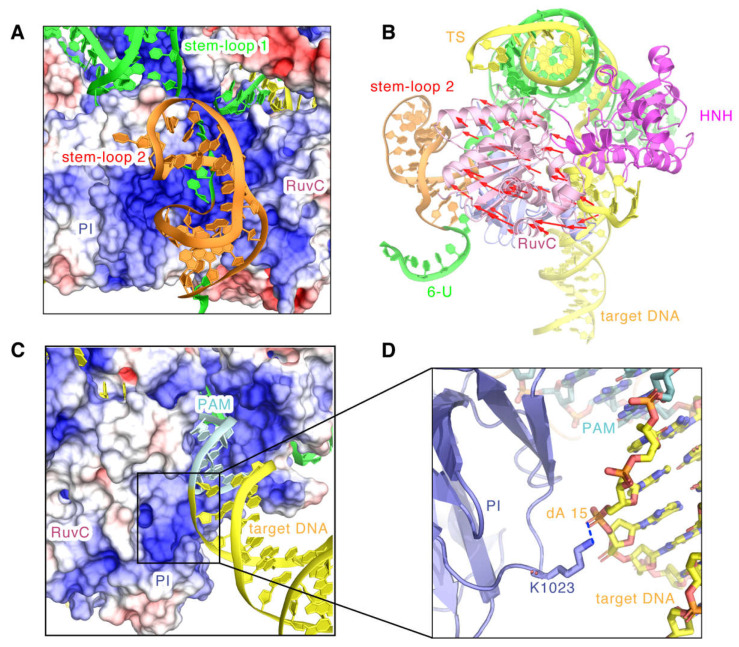
The topology of the stem-loop 2 and the post-PAM target DNA. (**A**) Electrostatic surface potential of SaCas9 bound to the stem-loop 2. The stem-loop 2 is represented by the cartoon in orange. (**B**) The movements of the RuvC domain towards the stem-loop 2. Red arrows represent the moving directions and distances. (**C**) Electrostatic surface potential of SaCas9 bound to the target DNA base downstream of the PAM. (**D**) The close contacts of the target DNA strand base downstream of the PAM with the PI domain. The target DNA bases downstream of the PAM are represented in yellow.

**Figure 8 ijms-24-01204-f008:**
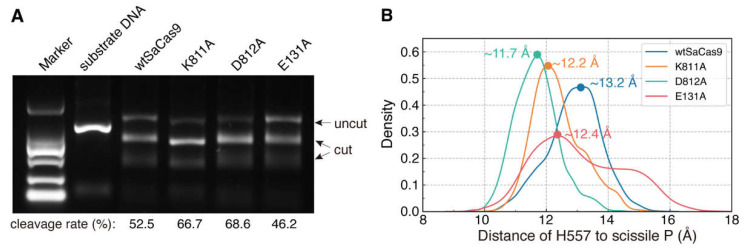
The effects of the REC and WED domains on the DNA cleavage by the HNH domain. (**A**) The cleavage activity of SaCas9 and its mutants (measured by ImageJ). (**B**) MD distance distributions between the catalytic residue H557 and the scissile P (D_557-P_) of SaCas9 and its mutants.

## Data Availability

The cryo-EM structure and the full-length model of the SaCas9-sgRNA-DNA complex have been deposited in the Electron Microscopy Data Bank (https://www.ebi.ac.uk/emdb/) under the accession code EMD-32104 and in the Protein Data Bank (https://www.rcsb.org) under the accession code 7VW3.

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
