# Peer review of "Full-Length Model of SaCas9-sgRNA-DNA Complex in Cleavage State"

_ijms, 2023, doi:10.3390/ijms24021204_

Round 1
Reviewer 1 Report
In this article entitled “Full-length model of SaCas9-sgRNA-DNA complex in cleavage State”, Du et al., performed a systemic first determine, via CryoEM, the complete active complex structure of SaCas9 which includes SaCas9, 104-nt sgRNA, 40-nt target DNA strand 28-nt non-target DNA strand. They observed that HNH nuclease domain binds to the cleavage site of the target DNA strand, and closely interacts with the WED and REC domains of SaCas9. The authors also performed site directed mutagenesis and molecular dynamics to confirm this interaction. This study is very systematic, well designed and will help the field move forward. The manuscript is also well-written.
Minor Concerns:
1) It would be nice if domain architecture of SaCas9 is explained in the introduction section to understand the results part of the manuscript better.
2) There are several grammatical mistakes throughout the manuscript including methods section, that should be proof-read carefully and fixed.
3) In the figure legends, the information about color coding should be included.
4) Some spellings should be corrected; Line 135- Except in place of expect. Please fix “Bond angles” in Table S2
Reviewer 2 Report
Du et al. have provided a nice study investigating the cryo-electron microscopy structure of the SaCas9-sgRNA-DNA ternary complex. The necessary information is provided in the manuscript. However, there are a few points that if considered will increase the value of the manuscript and may be readability.
-Does the manuscript possess a novelty? If yes, it will be good to add the novelty in the abstract. See it is already written in the introduction.
- Please elaborate on the discussion in light of the most current studies.
-Add a conclusion to the manuscript explaining the future prospect of the study in detail.
I do believe that the manuscript can be accepted once the authors address the mentioned points and enrich the manuscript with crucial information.
Round 2
Reviewer 2 Report
-